# Experimental Static Cold Storage of the Rat Uterus: Protective Effects of Relaxin- or Erythropoietin-Supplemented HTK-N Solutions

**DOI:** 10.3390/biomedicines10112730

**Published:** 2022-10-28

**Authors:** Lina Jakubauskiene, Matas Jakubauskas, Gintare Razanskiene, Bettina Leber, Diana Ramasauskaite, Kestutis Strupas, Philipp Stiegler, Peter Schemmer

**Affiliations:** 1General, Visceral and Transplant Surgery, Department of Surgery, Medical University of Graz, Auenbruggerplatz 2, 8036 Graz, Austria; 2Faculty of Medicine, Vilnius University, M. K. Ciurlionio Street 21, 03101 Vilnius, Lithuania; 3National Centre of Pathology, Affiliate of Vilnius University Hospital Santaros Klinikos, P. Baublio Street 5, 08406 Vilnius, Lithuania

**Keywords:** relaxin, erythropoietin, cold storage, animal model, uterus

## Abstract

Uterus transplantation (UTx) is the only treatment method for women with absolute uterine infertility. Currently, the number of grafts retrieved from deceased donors is increasing; hence, prolonged cold ischemia time is inevitable. Thus, this study was designed to assess the effect of the novel relaxin (RLN)- or erythropoietin (EPO)-supplemented Custodiol-N (HTK-N) solutions in an experimental uterus static cold storage (SCS) model. A total of 15 Sprague Dawley rats were used. Uterus horns were randomly assigned into three groups (*n* = 10/group). SCS was performed by keeping samples at 4 °C in HTK-N solution without or with different additives: 10 IU/mL EPO or 20 nM RLN. Tissue samples were taken after 8 and 24 h of preservation. Uterine tissue histology, and biochemical and immunohistochemical markers were analyzed. No significant differences in SCS-induced tissue damage were observed between groups after 8 h of preservation. Uterine tissue histology, MDA, SOD levels and the TUNEL-positive cell number showed severe damage in HTK-N without additives after 24 h of preservation. This damage was significantly attenuated by adding RLN to the preservation solution. EPO showed no favorable effect. Our study shows that RLN as an additive to an HTK-N solution can serve as an effective uterine tissue preservative in the uterus SCS setting.

## 1. Introduction

Uterus transplantation (UTx) is gaining interest as being the only treatment method for women with absolute uterine infertility. It affects 1 in 500 women of reproductive age and results either from congenital or acquired causes [1]. Uterine agenesis in Mayer–Rokitansky–Küster–Hauser syndrome is one of the congenital causes, which is relatively rare with a prevalence of 1 in 4500 women, while hysterectomy is the second most common female surgery mainly performed due to large myomas, peripartum bleeding, postpartum sepsis or malignancies [2]. UTx can serve as an alternative to adoption or surrogacy and enable a woman to have a childbirth experience [3].

The first UTx was performed in Saudi Arabia in 2000 [4]. Unfortunately, the transplanted graft had to be removed due to thrombosis three months after UTx. In 2014, a Swedish team led by Brännström successfully performed UTx, which resulted in the first live birth [2]. Since then, more than 40 UTxs have been successfully performed including transplantations from deceased donors [1,5]. There is an interest in improving protocols for UTx from deceased donors as this eliminates the risk of donor morbidity and mortality [6]. Furthermore, although the surgical procedure for performing UTx from a living or deceased donor is the same, longer graft vascular pedicles can be dissected from deceased donors, which further increases the probability of the success of the transplant [6,7]. For these reasons, research into the factors influencing the success of UTx from deceased donors is needed.

One of the primary factors influencing the ability to perform UTx from deceased donors is the time during which the ex situ uterus is stored. Static cold storage (SCS) has been the standard method for solid organ preservation since the 1960s [8]. The use of this method relies on reduced temperature, which causes the decrease in cell metabolic activity to the minimum required for survival and ultimately prolongs graft viability [9,10]. The susceptibility to SCS varies by organ and is shortest for the heart (4–6 h), whereas the kidney and liver can successfully survive cold ischemia for up to 44 h [11]. Previously published studies report uterine tolerance to cold ischemia for at least 6 h with maintained contractile function and histological characteristics [12]. Custodiol-N is a novel preservative solution developed after base Custodiol^®^ solution was supplemented with glycine, alanine, iron chelators and sucrose instead of mannitol [13]. Custodiol-N showed its superiority to Custodiol^®^ in a uterus SCS model likely by reducing tissue edema and inhibiting oxidative stress [14].

In recent years, a concerted scientific effort has been dedicated to finding ways to reduce tissue damage following SCS [15,16]. In particular, studies addressing whether the additional substances could be employed to minimize tissue damage during SCS.

Relaxin (RLN) is a peptide hormone, which belongs to the insulin family of structurally-related molecules [17]. The RLN receptor (RXFP1) is expressed on many organs, such as the kidney, heart, brain, arteries, lungs and even blood cells [18]. It plays an important role during the early stages of pregnancy and is highly expressed in reproductive tissues such as the ovaries, uterus (both endometrium and smooth muscles) and placenta. This hormone has already been studied in cell culture and animal models and demonstrated its antiinflammatory, antifibrotic, cytoprotective and antioxidant properties as well as a role in hemodynamic changes [19,20,21,22]. Given its central role, it is also under investigation in transplantation research [23].

Erythropoietin (EPO) is an endogenous hormone, produced by interstitial fibroblasts in the kidney or fetal liver under hypoxic conditions [24]. It is used for anemia treatment in people who need dialysis or have cancer. EPO receptors were found outside the hemopoietic system, such as in the uterus’ epithelial and stromal cells [25,26]. EPO showed its antioxidant effect by reducing lipid peroxidation and increasing superoxide dismutase (SOD) activity [27]. Moreover, preservation solutions supplemented with EPO reduced liver damage during SCS [28].

This study was designed to investigate the protective effect of RLN or EPO in a rat uterus SCS model and to examine morphological and biochemical changes in rat uterine tissue over different SCS time periods.

## 2. Materials and Methods

### 2.1. Animals

A total of 15 female (12 weeks old, weighing 250–330 g) Sprague Dawley rats (Janvier Labs, Le Genest-Saint-Isle, France) were used for this study. All rats were housed in the Division for Biomedical Research at the Medical University, Graz, and kept under controlled environment with a 12-h light/dark cycle, 22 ± 1 °C temperature and unlimited access to fresh water and standard rat chow. Before conducting the experiments, rats were acclimatized to the new surroundings for 7 days. The study was approved by the Austrian Ministry for Science, Research and Economy on the 17th of September 2019 (approval number: BMBWF-66.010/0154-V/3b/2019) and conducted according to the 3R guidelines.

### 2.2. Experimental Design

A total of 15 rats were used in an experimental SCS model. The initial anesthesia was achieved with 2% 2 L/min isoflurane (Piramal-Isoflurane, Piramal Critical Care Deutschland GmbH, Hallbergmoos, Germany). After the induction, anesthetic agent mixture consisting of 0.15 g/kg medetomidine (Domitor, Orion Pharma, Vienna, Austria), 2 mg/kg midazolam (Midazolam, Accord Healthcare GmbH, Munich, Germany) and 5 μg/kg fentanyl (Fentanyl-Hameln, Hameln, Germany) was injected intramuscularly. After midline laparotomy, the uterus was procured, and each uterus horn was randomly assigned into one of three groups (*n* = 10/group). Uterus SCS was performed by using Custodiol-N (HTK-N) solution in different compositions: Group HTKN (HTK-N + no additives), Group EPO (HTK-N + 10 IU/mL recombinant human erythropoietin (Erypo; Janssen-Cilag GmbH, Germany)) and Group RLN (HTK-N + 20 nM synthetic human relaxin-2 (Relaxera, Bensheim, Germany)). The concentrations for experimental substances were chosen according to previously published studies [28,29]. Uterine horns were gently flushed with the respective experimental preservation solution, immediately packaged into the same solution and placed on ice at 4 °C. Tissue samples were taken after 8 and 24 h of preservation. These timepoints were chosen based on our research group published data [14]. One part of the sample was flash-frozen in liquid nitrogen and stored at −80 °C for biochemical analysis and the other part was fixed in 4% formalin for histological analysis.

### 2.3. Tissue Morphological Evaluation

Uterine tissue was fixed in 4% formalin and embedded in paraffin and sections of 2 μm were stained with hematoxylin and eosin (H&E). Slides were evaluated by a blinded pathologist using an adapted scoring system for uterus SCS damage and analyzing the whole section at 100× magnification (Table 1) [14].

### 2.4. Biochemical Analyses

Two biochemical markers were chosen to evaluate the tissue oxidative stress. Malondialdehyde (MDA) is the final product of polyunsaturated fatty acids’ peroxidation in the cells and SOD acts as an agent against reactive oxygen species [30,31].

Frozen tissue samples were homogenized with ceramic beads in a phosphate-buffered solution (PBS) or Lysis buffer using the MagNA Lyser instrument (Roche Life Science, Mannheim, Germany). After homogenization, samples were centrifuged at 10,000 rpm for 10 min at 4 °C. The following assay kits, according to the manufacturer’s manuals, were performed using the collected supernatant: Pierce™ BCA Protein Assay Kit (catalog number: 23225, Thermo Fisher Scientific, Waltham, MA, USA), Lipid peroxidation (MDA) assay kit (catalog number: ab118970, Abcam, Cambridge, United Kingdom), Superoxide Dismutase Colorimetric Activity Kit (catalog number: EIASODC, Invitrogen, Thermo Fisher Scientific, Waltham, MA, USA). Tissue MDA and SOD levels were measured and expressed per milligram (mg) of tissue protein.

### 2.5. Terminal Deoxynucleotidyl Transferase-Mediated dUTP Nick-End Labeling (TUNEL) Assay

Cell apoptosis was evaluated using TUNEL Assay Kit—HRP-DAB (catalog number: Ab206386, Abcam, Cambridge, United Kingdom). The staining was performed according to the manufacturer’s manual.

Stained sections were scanned and positive cells in whole section were counted using QuPath software (open source software for Quantitative Pathology, version 0.2.0) [32]. The number of positive cells was expressed as the percentage of stained cells of total nuclei.

### 2.6. Statistical Analysis

Statistical analysis was performed using SPSS 23.0 (SPSS Inc., Chicago, IL, USA) and GraphPad Prism 9 (GraphPad Software, La Jolla, CA, USA). Variable distribution was determined using the Shapiro–Wilk test. As most of the continuous variables were non-normally distributed, the Kruskal–Wallis test and Dunn’s post hoc test were used for analysis. Data are reported as median and quartiles (Q1; Q3). Statistical significance was considered when *p* < 0.05.

## 3. Results

### 3.1. Tissue Biochemical Analysis

After 8 h of organ preservation, no significant differences in tissue MDA concentration and SOD activity were observed between the groups (Figure 1A,B). However, after 24 h of preservation, MDA levels increased in all groups compared to 8 h and differences between the groups after 24 h became significantly different. MDA concentration was highest in the HTKN group with no additives. MDA concentration in samples of the RLN group was significantly lower compared to samples in the HTKN group. Adding RLN to the preservative solution inhibited lipid peroxidation as samples after 24 h presented with similar MDA levels when compared to the 8 h group. EPO did not have a significant effect on MDA concentration after 24 h of SCS.

Moreover, differences in SOD activity between the groups became evident only after 24 h of SCS. The lowest SOD activity was observed in the HTK-N without additives group. In samples kept in the HTK-N solution with RLN, SOD activity remained significantly higher and was even similar compared to samples measured 8 h after preservation. No significant differences were observed in samples kept in the HTK-N solution supplemented with EPO.

### 3.2. Histology and Immunohistochemistry

The histological examination of uterine tissue showed no differences between groups after 8 h of preservation (Figure 1C and Figure 2). Samples of all groups presented with mild tissue edema and minor changes in the epithelium (Figure 2). In contrast, preservation of 24 h led to significant uterine tissue injury. Although no tissue necrosis was observed across any group, samples exhibited signs of severe edema, smooth muscle contraction and changes in the endometrium. According to our adapted scoring system, samples kept in HTK-N with no additives or with EPO had a higher tissue injury score compared to samples kept in the HTK-N solution supplemented with RLN.

TUNEL staining revealed a low apoptotic activity ranging from 0.14 to 0.47% in all experimental groups after 8 h of preservation (Figure 1D and Figure 2). After 24 h of preservation, the TUNEL-positive cell numbers were significantly higher compared to those after 8 h, the highest being in the HTK-N solution without additives (3.89% (IQR 2.02; 4.97)). The TUNEL-positive cell number was significantly decreased only in samples when the HTK-N solution was supplemented with RLN. Apoptotic activity in samples preserved in HTK-N + RLN did not increase between the 8 h and 24 h timepoints. No favorable effect of EPO as an additive was observed.

## 4. Discussion

Here we investigated whether RLN or EPO could alleviate the negative impact of SCS on uterus tissue in a rat uterus SCS model. The results of this study revealed that RLN could serve as an effective additive to an HTK-N preservation solution by reducing lipid peroxidation, oxidative stress and apoptosis and preserving uterine tissue morphology.

Many animal studies have already been conducted to assess uterus myometrial tissue tolerance to cold ischemia. From recent studies it is known that the murine uterus can tolerate cold ischemia for up to 24 h and still obtain a pregnancy [33]. Our study demonstrates that cold ischemia lasting up to 24 h has a negative impact on the uterine tissue resulting in biochemical and histological changes, but the damage can be diminished by adding RLN to the preservation solution. These findings are important as the uterus is not a vital organ and therefore is procured lastly from a brain-dead donor, resulting in an extended ischemic time [7].

SCS induces several pathophysiological mechanisms, which ultimately result in tissue damage. It impairs the activity of Na+/K+ ATPase, which causes cell swelling, depletes ATP and shifts cell metabolism to an anaerobic state; furthermore, one of the most important damage pathways is the formation of excess reactive oxygen species [10]. Therefore, in this study we decided to use the particular uterine tissue lesion indicators (lipid peroxidation (MDA) and antioxidative (SOD) markers, impairment of tissue morphology, and apoptotic cell number) as their changes are associated with posttransplant outcomes [34,35].

Various preservation solutions, as well as different additives, have already been used in the experimental and clinical practice of UTx. The most studied preservation solutions are Celsior^®^, HTK (Custodiol^®^), University of Wisconsin solution, Perfadex^®^ and IGF-1^®^ [1,36,37,38,39,40,41,42,43]. Some studies revealed that adjuvants such as L-carnitine and Iloprost to an HTK solution significantly attenuate uterine tissue alteration after 24 h of cold ischemia when compared to preservation in a plain HTK solution [15,16]. Currently, there is no consensus concerning the best storage fluid for uterus SCS. However, in a previous study, our group found that Custodiol-N is superior to Custodiol in a uterus SCS model most likely by inhibiting oxidative stress and reducing tissue edema [14]. Therefore, we investigated the possibility to further improve HTK-N solution in the setting of uterine SCS.

RLN is one of the protective agents that is already being widely studied in preclinical kidney, lung, heart and liver transplantation research [23,29,44,45,46,47,48,49,50,51,52,53]; however, to our knowledge, this has not yet been applied in a uterine transplant. RLN acts through different target molecules and pathways of which some are organ related [54]. It reduces expression of intracellular adhesion molecule 1, induces expression of Notch1 in macrophages and reduces the adhesion of neutrophils through the increased synthesis of nitric oxide [23,45]. In hepatocytes, it acts via glucocorticoid receptors and inhibits the release of cytochrome c from mitochondria showing antiapoptotic properties [51]. Moreover, RLN reduces vasoconstriction via the inhibition of endothelin 1 production and increases vasodilation through the increased production of nitric oxide [47].

Our study results revealed that RLN added to an HTK-N solution reduces MDA concentration and preserves SOD activity in rat uterine horns after SCS. Moreover, this translated into the histological findings such as reduced morphological damage and the inhibition of cellular apoptosis. Our findings are in line with the reported results from a similar study by Boehnhart et al. in which the effects of RLN were tested on liver tissue during SCS [55]. This study reported that RLN managed to sustain lower MDA and myeloperoxidase levels in hepatocytes.

The second preservative additive used in our study was EPO. Several of its protective properties were demonstrated in a variety of studies [27,56,57]. We chose this substance as, currently, there are no studies analyzing its properties on uterus SCS. Unfortunately, the addition of EPO to the HTK-N solution at our tested dose did not yield any additional protective properties to the uterine tissue in our study.

There are several limitations of our study that should be addressed. First of all, this is an animal study conducted in an experimental rat model. Even though these experimental results are promising, it remains unclear if they will fully generalize to human tissues and organs. Furthermore, our study design mainly focused on the uterine tissue’s morphologic and biochemical changes, and we did not examine the possible functional changes, which may be crucial for a successful uterus transplantation. Lastly, the human uterus is much thicker than the rat uterus, making it even more prone to ischemia. This limits the translational value of the study.

## 5. Conclusions

In conclusion, our study shows that RLN as an additive to an HTK-N solution can serve as an effective uterine tissue preservative in a uterus SCS setting. Further studies are warranted for introducing its usage in clinical practice.

## Figures and Tables

**Figure 1 biomedicines-10-02730-f001:**
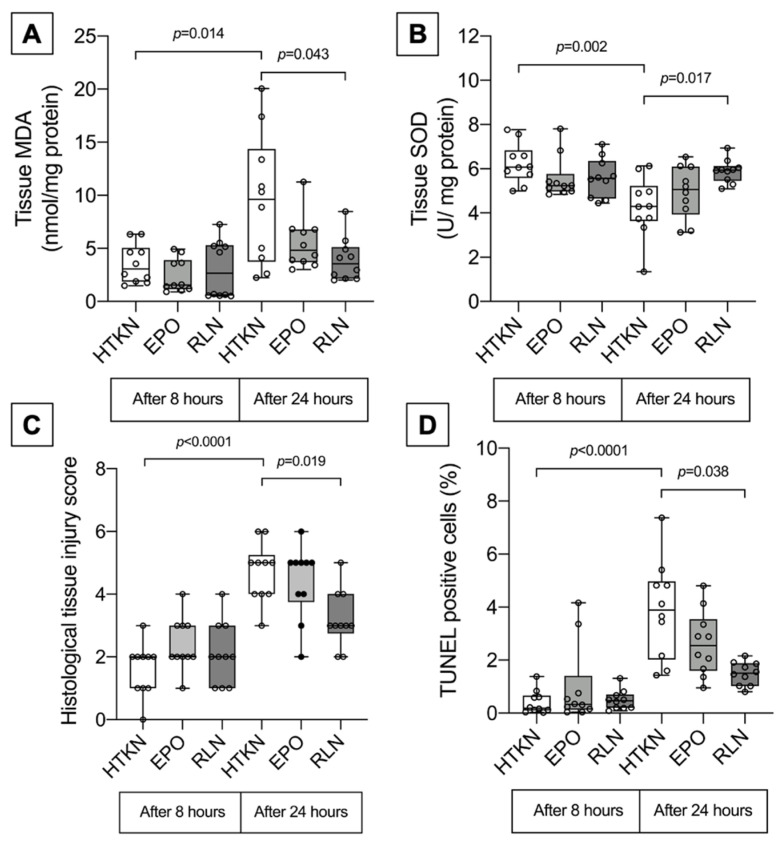
Uterine tissue lesion markers after 8 and 24 h of SCS. (**A**) uterine tissue MDA concentration, (**B**) uterine tissue SOD concentration, (**C**) histological uterine tissue injury score, (**D**) TUNEL positive cell count. HTK-N—Custodiol -N, EPO—erythropoietin, RLN—relaxin, MDA—malondialdehyde, SOD—superoxide dismutase, TUNEL—terminal deoxynucleotidyl transferase-mediated dUTP nick-end labeling. Data presented as median (Q1; Q3).

**Figure 2 biomedicines-10-02730-f002:**
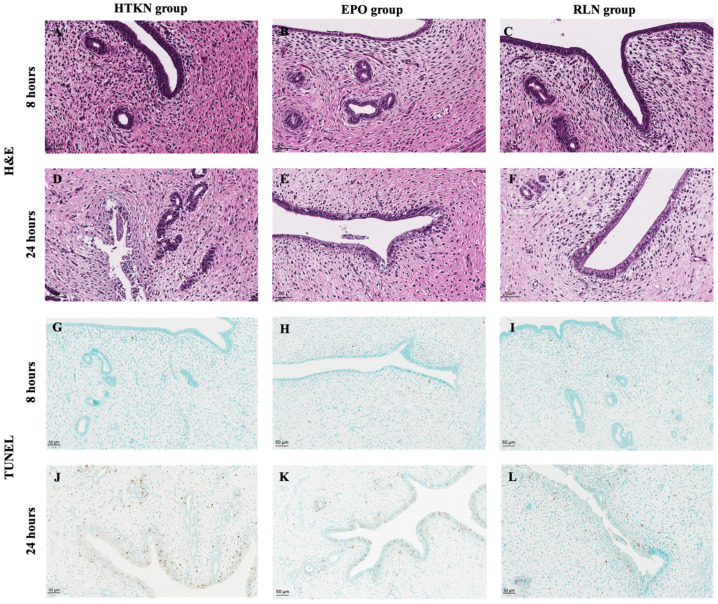
Micrographs of histological changes (**A**–**F**) and TUNEL-positive cells (**G**–**L**) in uterine tissue after 8 h and 24 h of preservation. (**A**–**C**) Samples after 8 h of preservation presented with mild tissue edema and minor changes in the epithelium. (**D**,**E**) Preservation of 24 h in HTK-N without additives or with EPO caused signs of tissue severe edema, smooth muscle contraction and changes in the endometrium. (**F**) Preservation of 24 h in HTK-N solution with RLN, samples presented with tissue edema, vacuolization of the endometrium and preserved integrity of surface epithelium. HTK-N—Custodiol-N, EPO—erythropoietin, RLN—relaxin, H&E—hematoxylin and eosin, TUNEL—terminal deoxynucleotidyl transferase-mediated dUTP nick-end labeling.

**Table 1 biomedicines-10-02730-t001:** Light microscopy morphological evaluation scoring system for uterus SCS model.

	Score			
	0	1	2	3
Edema	Absent	<15%	15–30%	>30%
Necrosis	Absent	<15%	15–30%	>30%
Smooth muscle contraction	Absent	Present		
Impaired integrity of surface epithelium	Absent	Present		
Disruption of epithelial cells	Absent	Present		
Endometrial loss of cells	Absent	Present		

Note Percentages are calculated as (surface of the affected area/surface of the whole section) × 100.

## Data Availability

All data relevant to the study are included in the article.

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
