# Peer review of "Experimental Static Cold Storage of the Rat Uterus: Protective Effects of Relaxin- or Erythropoietin-Supplemented HTK-N Solutions"

_biomedicines, 2022, doi:10.3390/biomedicines10112730_

Round 1

Reviewer 1 Report

Jakubauskiene et al describe a comparison of cryoprotectant regimes to facilitate heterologous uterine transplantation. Because this is a rapidly advancing field, new approaches aiming to increase surgical yield are welcome. This experimental work contributes to that effort and should be given serious consideration for publication. Only a few issues should be addressed first, as outlined below:

Major points

In the methods section, the scoring system used for post-operative tissue assessment is clearly presented and the Table is helpful. However, the number of fields examined microscopically is not stated. Surely the blinded pathology expert views multiple fields before scoring (see lines 120-121)? For example, the authors could indicate a composite score was used based on an average taken over at least X fields at defined magnification etc.

Minor issues

Title: check capitalization in rat uterus

Introduction: While RKH is an unusual reason for uterine agenesis, surgical removal of the uterus (hysterectomy) is the second most common female surgery and is performed for myomas, dysfunctional bleeding, endometriosis, pelvic pain, malignancy etc …

Line 40: …enable a woman to have a childbirth experience.

Line 53:   ex situ    

Line 54: since the 1960’s

Line 96:  Science, Research & Economy (?)

Lines 234-5:  however, to our knowledge this has not yet been applied in a uterine transplant perioperative context.

Line 256-7:  even though these experimental results are promising, it remains unclear if these will fully generalize to human tissues and organs.

Author Response

Reviewer 1, Comment 1: In the methods section, the scoring system used for post-operative tissue assessment is clearly presented and the Table is helpful. However, the number of fields examined microscopically is not stated. Surely the blinded pathology expert views multiple fields before scoring (see lines 120-121)? For example, the authors could indicate a composite score was used based on an average taken over at least X fields at defined magnification etc.

Reply: Thank you for the comment. The scoring system that is detailed in Table 1 relies on either the area or the presence or absence of a pathological criteria and they are evaluated from different parts of the uterus section (epithelium, smooth muscles etc.). In order to use this scoring system, the pathologist has to evaluate the whole section and the evaluation based on an average taken from X fields is not possible. To clarify the evaluation process to the reader we have additionally changed the sentence in the methods section to “Slides were evaluated by a blinded pathologist using an adapted scoring system for uterus SCS damage and analyzing the whole section at 100x magnification (Table 1)”.

Reviewer 1, Comment 2: Title: check capitalization in rat uterus.

Reply: Thank you, we have corrected this mistake.

Reviewer 1, Comment 3: Introduction: While RKH is an unusual reason for uterine agenesis, surgical removal of the uterus (hysterectomy) is the second most common female surgery and is performed for myomas, dysfunctional bleeding, endometriosis, pelvic pain, malignancy etc.

Reply: We have made a change to the manuscript according to your comment.

Reviewer 1, Comment 4: Line 40: …enable a woman to have a childbirth experience.

Reply: We have made a change to the manuscript according to your comment.

Reviewer 1, Comment 5: Line 53: ex situ Line 54: since the 1960’s Line 96:  Science, Research & Economy (?)

Reply: We have made a change to the manuscript according to your comment.

Reviewer 1, Comment 6:

Lines 234-5: however, to our knowledge this has not yet been applied in a uterine transplant perioperative context.

Reply: We have made a change to the manuscript according to your comment.

Reviewer 1, Comment 7:

Line 256-7: even though these experimental results are promising, it remains unclear if these will fully generalize to human tissues and organs.

Reply: We have made a change to the manuscript according to your comment.

Reviewer 2 Report

The manuscript by Jakubauskiene et al. investigated the effects of relaxin or erythropoietin supplemented Custodiol-N (HTK-N) solution in an experimental uterus static cold storage model. The model is sound, and the manuscript is well written. However, I have some concerns that should be addressed. I provide below some comments/suggestions that should be addressed before further consideration. 

MAJOR:

Authors added human EPO and RLN to the control media. Is there literature/studies supporting that these human proteins can bind and activate their respective receptors in the rat?

Is there an explanation why authors decided to present the data as median and quartiles?

Regarding statistical analyses, it is not clear to this reviewer how the effects of time (8 vs 24 h) and group (HTKN vs EPO vs RLN) were analyzed since authors state that a one-way ANOVA was performed. Ideally, a two-way ANOVA should be performed including time, group, and time*group interaction as factors. Moreover, a paired T-test could be performed since same uterine horns were sampled at the 8 and 24 h time-points. 

Regarding pairwise comparisons (Tukey test), were groups compared to all other groups?

In Fig 1C, it is not clear how the tissue injury score (0 to 6) was calculated from the material and methods section. Could authors expand on this? The same applies to TUNEL positive cells (%) calculations (Fig. 1D).

Where are data using scores from Table 1 represented?

In Fig 2, all groups/time-points should be shown for both H&E and TUNEL. 

Authors state that “According to our adapted scoring system, samples kept in HTK-N with no additives or with EPO had a higher tissue injury score compared to samples kept in HTK-N solution supplemented with RLN (lines 180-181). Where are the data supporting this?

Authors should add to the study limitations the fact that the human uterus is much thicker that the rat uterus, making it even more prone to ischemia. This limits the translational value of the study.

MINOR:

There should be a space between the number and unit (e. g., 8 h instead of 8h). Please correct throughout the manuscript. 

Lines 107-109: For clarity and conciseness, I suggest groups be named as data are presented in the graphs (HTKN, EPO, and RLN) instead of group I, II and III. 

Line 146: “Normally distributed data was analyzed” should be “Normally distributed data were analyzed”.

Lines 147-148: “Non-normally distributed data was evaluated” should be “Non-normally distributed data were evaluated”

Line 150: “Data is reported” should be corrected to “Data are reported”.

Lines 158-159: “Samples in HTK-N+RLN group presented with significantly lower MDA” sounds off. Please re-word. 

Line 166: “8 hours” should be “8 h”. Please correct elsewhere in the manuscript for consistency. 

Lines 178-179: “Although no tissue necrosis was observed across all groups” should be “Although no tissue necrosis was observed across any group”.

Author Response

Reviewer 2, Comment 1: Authors added human EPO and RLN to the control media. Are there literature/studies supporting that these human proteins can bind and activate their respective receptors in the rat?

Reply: The same Relaxin family peptide receptor 1 is expressed in both human and rat uterus (Bathgate R a. D, Halls ML, van der Westhuizen ET, Callander GE, Kocan M, Summers RJ. Relaxin family peptides and their receptors. Physiol Rev. 2013 Jan;93(1):405–80.). Additionally, the same Erythropoietin receptors are found within mammals (Halvorsen S, Bechensteen A. Physiology of erythropoietin during mammalian development. Acta Paediatrica. 2007 Jan 2;91:17–26.).

Reviewer 2, Comment 2: Is there an explanation why authors decided to present the data as median and quartiles?

Reply: Most of the data in this article was not normally distributed thus to keep consistence throughout the article we chose to report median and quartiles.

Reviewer 2, Comment 3: Regarding statistical analyses, it is not clear to this reviewer how the effects of time (8 vs 24 h) and group (HTKN vs EPO vs RLN) were analyzed since authors state that a one-way ANOVA was performed. Ideally, a two-way ANOVA should be performed including time, group, and time*group interaction as factors. Moreover, a paired T-test could be performed since same uterine horns were sampled at the 8 and 24 h time-points.

Reply: The main purpose of this study was to evaluate the effects of the substances used in reducing tissue damage. Basically, the 8-hour timepoint was used as a baseline for tissue damage, also showing that the damage was relatively low thus the differences in used substances did not become apparent. The 24-hour timepoint eventually showed that with the increasing tissue damage the used additives (only RLN) had protective effect. With our statistical analysis we did not intend to show time interaction on tissue damage, thus we did not use the two-way ANOVA or paired T-test.

Reviewer 2, Comment 4: Regarding pairwise comparisons (Tukey test), were groups compared to all other groups?

Reply: During the revision of the manuscript, we have noticed that an error was left in the primary version of the manuscript that we used ANOVA as only Kruskal-Wallis with Dunn’s pairwise comparison test was used due to non-normally distributed data, we have amended this in the revised manuscript. Yes, all groups were compared to all other groups in the pairwise comparisons (Dunn’s test).

Reviewer 2, Comment 5: In Fig 1C, it is not clear how the tissue injury score (0 to 6) was calculated from the material and methods section. Could authors expand on this? The same applies to TUNEL positive cells (%) calculations (Fig. 1D).

Reply: The scoring system that is detailed in Table 1 relies on either the area or the presence or absence of a pathological criteria and they are evaluated from different parts of the uterus section (epithelium, smooth muscles etc.). In order to use this scoring system, the pathologist has to evaluate the whole section. To clarify the evaluation process to the reader we have additionally changed the sentence in the methods section to “Slides were evaluated by a blinded pathologist using an adapted scoring system for uterus SCS damage and analyzing the whole section at 100x magnification (Table 1)”. Additionally, we have improved the description on how TUNEL positive cells were counted: “Stained sections were scanned and positive cells in whole section were counted using QuPath software (open source software for Quantitative Pathology, version 0.2.0). The number of positive cells was expressed as the percentage of stained cells of total nuclei.“

Reviewer 2, Comment 6: Where are data using scores from Table 1 represented?

Reply: Thank you for the comment, the scoring system summarized data is presented in the figure 1C and also briefly mentioned in text in the results paragraph.

Reviewer 2, Comment 7: In Fig 2, all groups/time-points should be shown for both H&E and TUNEL.

Reply: Thank you for the comment, we have made a new figure 2 with all the groups/time-points and added it to the manuscript.

Reviewer 2, Comment 8: Authors state that “According to our adapted scoring system, samples kept in HTK-N with no additives or with EPO had a higher tissue injury score compared to samples kept in HTK-N solution supplemented with RLN (lines 180-181). Where are the data supporting this?

Reply: Thank you for the comment, we are not exactly sure whether we correctly understood your question, but the scoring system data is presented in the figure 1C and also briefly mentioned in text in the results paragraph.

Reviewer 2, Comment 9: Authors should add to the study limitations the fact that the human uterus is much thicker that the rat uterus, making it even more prone to ischemia. This limits the translational value of the study.

Reply: Thank you for the insightful comment, we have added this to the study limitations section.

Reviewer 2, Comment 10: There should be a space between the number and unit (e. g., 8 h instead of 8h). Please correct throughout the manuscript.

Reply: We have made a change to the manuscript according to your comment.

Reviewer 2, Comment 11: Lines 107-109: For clarity and conciseness, I suggest groups be named as data are presented in the graphs (HTKN, EPO, and RLN) instead of group I, II and III.

Reply: We have made a change to the manuscript according to your comment.

Reviewer 2, Comment 12: Line 146: “Normally distributed data was analyzed” should be “Normally distributed data were analyzed”.

Reply: We have made a change to the manuscript according to your comment.

Reviewer 2, Comment 13: Lines 147-148: “Non-normally distributed data was evaluated” should be “non-normally distributed data were evaluated”

Reply: We have made a change to the manuscript according to your comment.

Reviewer 2, Comment 14: Line 150: “Data is reported” should be corrected to “Data are reported”.

Reply: We have made a change to the manuscript according to your comment.

Reviewer 2, Comment 15: Lines 158-159: “Samples in HTK-N+RLN group presented with significantly lower MDA” sounds off. Please re-word.

Reply: Thank you for the comment, we have rephrased the sentence.

Reviewer 2, Comment 16: Line 166: “8 hours” should be “8 h”. Please correct elsewhere in the manuscript for consistency.

Reply: We have made a change to the manuscript according to your comment.

Reviewer 2, Comment 17: Lines 178-179: “Although no tissue necrosis was observed across all groups” should be “Although no tissue necrosis was observed across any group”.

Reply: We have made a change to the manuscript according to your comment.